# The Mediation Effect of Coping Strategies between Personality and Blood Pressure in Pregnancy Complicated by Hypertension

**DOI:** 10.3390/healthcare10020341

**Published:** 2022-02-10

**Authors:** Sabrina Chapuis-de-Andrade, Carmen Moret-Tatay, Tatiana Quarti Irigaray, Ivan Carlos Ferreira Antonello, Bartira Ercília Pinheiro da Costa

**Affiliations:** 1Postgraduate Program in Medicine and Health Sciences, Pontifical Catholic University of Rio Grande do Sul, Porto Alegre 90619-900, Brazil; sabrinachapuis@gmail.com (S.C.-d.-A.); ivan.antonello@pucrs.br (I.C.F.A.); bart@pucrs.br (B.E.P.d.C.); 2Research Fellow from Coordination of Superior Level Staff Improvement—Brazil, CAPES Foundation, Brasília 70040-031, Brazil; 3Facultad de Psicología, Universidad Católica de Valencia San Vicente Mártir, 46001 Valencia, Spain; 4Dipartimento di Neuroscienze Salute Mentale e Organi di Senso (NESMOS), Università Sapienza di Roma, 00185 Rome, Italy; 5Postgraduate Program in Psychology, Pontifical Catholic University of Rio Grande do Sul, Porto Alegre 90619-900, Brazil; tatiana.irigaray@pucrs.br

**Keywords:** pregnancy, hypertension, pregnancy-induced, personality, blood pressure, coping

## Abstract

Background: Hypertension is the most common medical condition during pregnancy. Hypertensive disorders are associated with an increased risk of adverse outcomes for both mother and fetus. This study examined the role played by personality and coping strategies in relation to blood pressure levels during pregnancy. The specific goal was to study whether coping strategies can mediate the effect of personality in pregnant women with hypertension. Methods: A sample of 351 pregnant women was enlisted, encompassing 192 pregnancies complicated by hypertension. This is a cross-sectional study where personality traits were measured by the five-factor model and coping was evaluated by Jalowiec’s coping inventory scale. Results: Personality can partially predict systolic and diastolic blood pressure. Openness to experience trait is inversely correlated with systolic and diastolic blood pressure. Conversely, emotion-focused coping directly correlated with blood pressure levels. Systolic (β = −0.14; *p* < 0.05) and diastolic (β = −0.15; *p* < 0.05) blood pressure were also predicted by openness to experience. Conclusions: It is recommended to reinforce the development of coping strategies which focus more on the problem than on the emotion, avoiding detrimental effects of emotional coping in blood pressure levels during pregnancy.

## 1. Introduction

Hypertension is one of the most common causes of medical complications during pregnancy [1]. It affects 5 to 10% of pregnancies and is the main factor of maternal and fetal mortality and morbidity worldwide [2].

There are different classifications for hypertensive disorders in pregnancy. According to the National High Blood Pressure Education Program Working Group on High Blood Pressure in Pregnancy, they can be classified as follows: gestational hypertension, preeclampsia/eclampsia, chronic hypertension, and preeclampsia superimposed on chronic hypertension [3]. Gestational hypertension is characterized by high blood pressure (a systolic blood pressure greater than or equal to 140 mmHg or a diastolic blood pressure greater than or equal to 90 mmHg), normalized no later than the twelfth week after childbirth, without the presence of proteinuria. However, many women can progress to preeclampsia, which is determined by the combined presentation of high blood pressure, proteinuria, and organ dysfunction [4]. Some cases of preeclampsia can develop seizures, also known as eclampsia, a medical emergency. Chronic hypertension is traditionally diagnosed by high blood pressure before the twentieth week of gestation and does not disappear after week twelve postpartum. Women who are chronically hypertensive and present proteinuria after the twentieth week of gestation are considered to have preeclampsia superimposed on chronic hypertension [5].

Importantly, risk factors for hypertensive disorders in pregnancy may be of different etiologies. Nulliparity, maternal age, ethnicity, and higher blood pressures were identified as risk factors for developing hypertensive pregnancy disorders [4]. Even as most of these variables might be physical, several psychological variables should also be included [6]. More precisely, personality might play a crucial role as it has been considered an important predictor of physical and mental health [7,8].

Personality describes the consistent patterns of behavior, motivation, cognition, and emotion, which are unique for each person [9]. The assessment of personality can be determined by several methods. In this study, we use the five-factor model (Big Five), a suitable theoretical framework for understanding the nature of the relationship between personality and hypertensive disorders in pregnancy.

Regarding previous research into personality issues, authors found that subjects who achieve high scores on neuroticism and low on openness to experiences have the worst cardiovascular response to stress reactions [10]. Additionally, during pregnancy, neuroticism was associated with negative outcomes. Previous studies show that neuroticism was associated with the development of postpartum depression symptoms [11,12]. However, an openness personality trait can decrease the risk for postnatal depression symptomatology [8]. Mothers’ breastfeeding intentions were also negatively associated with neuroticism [13]. Moreover, women who have more traits of neuroticism and openness to experience, and who scored lower on consciousness and agreeableness, have a high risk for preterm labor [14].

It is known that subjects who score higher on the neuroticism trait tend to feel more negative emotions, such as nervousness, tension, insecurity, and worry [15]. Consequently, these individuals usually employ inefficient coping strategies. The development of coping strategies is crucial during pregnancy, due to several transformations taking place on physical and psychological levels. Therefore, coping strategies that women use to oppose problems can either contribute to the improvement of their health or further aggravate pregnancy disorders. This is especially true if the woman has a pregnancy complicated by hypertension, where they would need to deal with challenging adversities.

Coping is defined as a set of cognitive and behavioral strategies that subjects put to use in order to manage their stress levels [16]. Interestingly, there are different available strategies that deal with the same problem. However, coping can be classified into two main groups [17]: problem-focused (confrontive and supportant coping styles that usually respond actively to stressful situations) and emotion-focused (evasive, fatalistic, optimistic, emotive, palliative, and self-reliant coping styles that tend to deal more with feelings when they are in a stressful situation). The present study uses Jalowiec’s coping inventory to examine strategies for coping in pregnant women. This instrument consists of 60 items, with 8 styles of coping: confrontive (10 items), evasive (13), optimistic (9), fatalistic (4), emotive (5), palliative (7), supportant (5), and self-reliant (7) [17].

Additionally, previous studies have already shown that coping might, instead, serve as a life quality mediator [18]. A mediator is understood as a variable that may serve as an instrument through which one variable influence another [19]. Moreover, the mediation model recognizes that the subjects’ coping strategies are linked to their stress levels, transferring the effects to other underlying variables. In this study, we aim to understand if a specific coping strategy (emotion or problem focused) will lead to higher levels of blood pressure, regardless of personality.

Understanding human behavior is key to knowing how to approach health and disease issues in a singular and effective way. Although the literature is extensive on this topic, there are many questions that remain unclear concerning personality, coping, and blood pressure in the specific case of hypertension in pregnancy.

We hypothesized that some traits of personality can be associated with increased blood pressure during pregnancy. Consistent with previous reports, it was found that openness to experience was able to predict blood pressure, and this effect was larger for systolic pressure [20]. Since personality can be considered unalterable [9,21], other variables, such as coping strategies, might mediate its effect. For this proposition, the study puts forward a mediation model of coping, affecting the relationship of personality on blood pressure in pregnancy complicated by hypertension.

## 2. Materials and Methods

### 2.1. Participants

This cross-sectional study was conducted using an incidental sample made up of 351 pregnant women in the third trimester. Demographic and obstetric information were collected via medical records. Data included age, ethnicity, anthropometric data, education levels, employment and marital status, social support, parity, and previous pregnancy outcomes. Clinical features, such as laboratory tests (i.e., urinary protein and creatinine measurements), history of anemia, hypertension, or any other physical or psychological complication, were extracted from the medical records.

### 2.2. Instruments

#### 2.2.1. Mini Mental State Examination 

This instrument measures global cognitive functioning, such as temporal evaluations, space, attention, calculations, memory, language, and visual constructive capacity [22]. In this study, we used the Brazilian version [23]. Scores range between 0 (high cognitive impairment) and 30 points (better cognitive capacity). We included women achieving scores equal or superior to 27.

#### 2.2.2. The Five-Factor Model—Big Five 

It consists of 44 self-report items constructed to allow quick and efficient assessment of the 5 personality dimensions [24]. The instrument was adapted and validated for the Brazilian Portuguese language [25]. The participants answered a five-point Likert scale: strongly disagree, somewhat disagree, neither agree nor disagree, somewhat agree, and strongly agree. Mean scores were computed across the items of every scale separately. In this sample, the coefficients of internal reliability (General Cronbach’s α = 0.84) were: extraversion α = 0.88; agreeableness α = 0.75; conscientiousness α = 0.88; neuroticism α = 0.82; and openness to experience α = 0.70.

#### 2.2.3. Jalowiec’s Coping Inventory 

The Jalowiec’s Coping Inventory [26] was adapted and validated for the Brazilian Portuguese language [27], and it evaluates generic coping with concise descriptions of specific cognitive and behavioral strategies. Women answered how much they used each coping strategy to deal with or handle the stressor. Self-report ratings for each item were made on a Likert scale from 0 (never used) to 3 (often used). The coping style of participants was calculated in the following way: items marked in each question are summed up and divided by the number of items contained in the subscale, called the middle score; the mean score of each subscale is calculated by dividing and summing up the total mean scores; the coping style is defined by the highest score [28]. For this study, the internal consistencies for each scale were computed: confrontive (α = 0.79), supportant (α = 0.63), evasive (α = 0.81), fatalistic (α = 0.74), optimistic (α = 0.62), emotive (α = 0.79), palliative (α = 0.58), self-reliant (α = 0.47), and the global Cronbach’s alpha was 0.90. A Cronbach’s alpha of 0.77 was confirmed for problem-focused coping and an alpha of 0.87 was confirmed for emotion-focused coping.

#### 2.2.4. Procedure

This study comprises the Line of Research on Hypertension in Pregnancy. This work partly consists of a broader research effort, with previous results published but with different investigation objectives [29,30]. Thus, a survey was conducted from December 2016 to October 2018. Women filled out the questionnaires individually during their visit to a hospital in the southern region of Brazil, a specialized center for high-risk pregnant women care. Women were eligible if they were 18 years of age or older. Exclusion criteria included previous diagnosis of kidney disease, a history of diabetes, fetal malformation, and/or lack of information in the database. All women received a detailed explanation about the study. Written informed consent was obtained from all study participants. Completing the questionnaire took 40 min approximately, on average. The anonymity and confidentiality of data and voluntary participation were ensured, and all ethical standards were followed according to the National Research Council of Brazil (Resolution 466/2012) and the Code of Ethics of the World Association. The Local Institutional Review Board approved the study (Protocol No. 1.777.443-CEP).

### 2.3. Data Analysis

After descriptive and relational analyses based on Person’s correlation coefficient, a regression analysis was carried out to analyze the mediating role of coping. The statistical analysis was performed using SPSS 23 (IBM), and the mediational analysis used PROCESS macro [31]. This method measures the indirect effect that represents the impact of the mediator variable on the stipulated relation by a method of bootstrapping (10,000) with confidence intervals. More precisely, a regression coefficient (and associated t test) was first calculated on the mediational M variable (and its inherent a and b paths), the X independent variable on the dependent variable without the inclusion of moderator (c’ path), and the X independent variable on the dependent variable after the mediator was included (c path). Figure 1 depicts this analysis in terms of variables and paths. Of note, X is represented as the independent variable (personality), Y is the dependent variable (blood pressure), and M is the mediational variable (coping). Paths are represented by the letter a (the effect of X on M), b (the effect of M on Y), and c’ (the effect of X on Y), while c depicts the total effect.

## 3. Results

A total sample of 351 women participated in the study. According to prior hypertension in pregnancy diagnosis, women were classified as follows: (i) gestational hypertension (16.8%), (ii) preeclampsia syndrome (29.3%), (iii) chronic hypertension (8.5%), and (iv) control group encompassing healthy women with uncomplicated pregnancies (45.3%). With regards to the educational level, 60.7% had basic studies, 33.6% intermediate, and 5.7% superior ones. The proportion of Caucasian women in the control group was 24% and in the hypertensive group was 22%. The mean gestational age in the control group was 271.76 (SD = 12.77) days and in the hypertensive group 264.75 (SD = 19.52) days. In total, 32% of the control group were primiparous and 40% of women with hypertension were pregnant for the first time. The proportions of women that work or study in the control and hypertensive groups were 76% and 82%, respectively. Regarding social support, 0.9% of the control group and 1.3% of hypertensive women reported that they have no one to count on. Finally, more than 77.5% reported being married or in a relationship. Other demographic data are depicted in Table 1.

Secondly, the relationship between variables under study was examined. The Pearson coefficient was calculated among systolic and diastolic pressure, along with the Big Five traits of personality and coping (problem- and emotion-focused). Note that openness to experience trait and emotion-focused coping correlated with both blood pressures (Table 2).

On the other hand, a regression analysis on the prediction of blood pressure was carried out. Independent variables included the group (control and hypertensive) and the Big Five traits of personality and coping (problem and emotion focused. Diastolic blood pressure was predicted by openness to experience (β = −0.17; *p* < 0.05) and the group (β = −0.49; *p* < 0.001): F (11,339) = 13.35; *p* < 0.001; R^2^ = 0.30. Systolic blood pressure was also predicted by openness to experience (β = −0.15; *p* < 0.05), anxiety (β = −0.21; *p* < 0.001), and the group (β = −0.60; *p* < 0.001): F (11,339) = 23.97; *p* < 0.001; R^2^ = 0.42.

A mediational approach was carried out. The model was estimated in both strategies of coping simultaneously. In this way, the whole dataset was employed in order to have more data variability in the dependent variable blood pressure (in other words, the relationship between personality and blood pressure disappeared by splitting the groups). As expected, prediction of diastolic and systolic pressure, though openness to experience, reached the statistical significance level with a similar explained variance (*p* < 0.001; R^2^ = 0.22).

A similar pattern occurred for agreeableness for both pressures (*p* < 0.001; R^2^ = 0.14). Conscientiousness just predicted the Systolic pressure (*p* < 0.001; R^2^ = 0.13).

Figure 2 illustrates the simultaneous model and Table 3 depicts the confidence interval (CI) that was statistically significant with a confidence interval, excluding the zero value, at 95%. Once again, and as depicted in Table 3 and Table 4, two models were depicted according to diastolic or systolic blood pressure.

## 4. Discussion

In this study, we found that personality can partially predict systolic and diastolic blood pressure. Contrary to expectation, no effects were observed for neuroticism, when considered as a mediator of association between personality and blood pressure during pregnancy.

Neuroticism is known as a negative personality trait, understood as a tendency to feel anxiety, guilt, sadness, anger, and nervousness [9]. Moreover, it is associated with high reactivity, sensitivity to stress, and emotional instability [34]. Individuals presenting more neuroticism-related traits tend to have unrealistic ideas and usually face more difficulties in dealing with problems [15]. However, data establishing neuroticism as a predictor for disease and health status changes remain unclear. A prospective study found that higher neuroticism was strongly correlated to the risk of coronary heart disease [35]. In addition, authors of a different study found that neuroticism can be predictive of elevated blood pressure [36]. On the other hand, a large study that encompassed data from 76,150 participants failed to find an association between neuroticism traits and elevated mortality risk [37].

Our data shows that the openness to experience trait is inversely correlated with systolic and diastolic blood pressure. On the other hand, emotion-focused coping directly correlated with blood pressure levels. Regarding personality, previous studies have also found similar results. Authors found that personality, especially openness, could moderate the association between stressful events and blood pressure [20]. Other studies also found that openness was associated with both systolic and diastolic blood pressure responsivity [38]. Individuals who achieve high scores in openness to experience tend to be more creative, enjoy intellectual pursuits, ponder ideas, and seek new experiences and challenges in life [9]. This might explain why even women with pregnancy hypertension have healthier blood pressure levels than some control subjects. We hypothesized that even though they are hypertensive, they are more open to face this situation. Specifically, they might consider this period as a challenge to be surmounted and act more resourcefully when doing so. It is feasible to propose that, on a physiological level, the body presents a more adequate response to cardiovascular stress [20]. Moreover, blood pressure can be influenced since subjects with more openness to experience have a more adaptive profile. This is especially important in pregnant women as they experience many physical and psychological changes.

As highlighted by other authors, openness to experience has a positive correlation with cognitive ability and adaptive behaviors [39]. These aspects are of interest in the context of pregnancy, due to the fact that it is known that multiple risk factors may need to be modified in order to affect blood pressure—i.e., sedentary lifestyle, smoking, and improper feeding [40]. Therefore, women who achieve high scores in openness to experience may have more cognitive ability than others, resulting in a better repertoire of adaptive behaviors. Thus, they may have a healthier lifestyle and can control their blood pressure levels more effectively.

Indeed, conscientiousness-related traits, including aspects related to duty, planning, and organization [9], were not significant in this sample. It is known that people who exhibit higher levels of conscientiousness tend to be more organized and display more adherence to rules, i.e., by following medical prescriptions. Authors found a 1.4 times higher risk of death in subjects with lower expression of conscientiousness [37]. In the context of pregnancy hypertension, it is essential that women follow all recommendations to minimize complications resulting from this disorder, as well as controlling blood pressure.

Additionally, extraversion and agreeableness traits show no effects as mediators of associations between personality and blood pressure in pregnancy. Extraversion is related to enthusiasm, interest, and expansiveness characteristics, whereas agreeableness is linked to altruism and kindness [9]. Despite the differences between these two broad domains, where extraversion might be associated with the relationship between the external world and agreeableness with interpersonal tendencies, both are important to health outcomes. It has been reported that burnout syndrome in nursing is negatively associated with extraversion and agreeableness [41]. Considering that mental health is just as important as physical health, especially in pregnancy, it was expected that women presented characteristics such as sensitivity to positive emotions.

On the other hand, we found that those participants in the study who had coping focused on emotion also had higher average blood pressure levels. The emotion-focused coping strategy is understood as passive coping or avoidance, which is effective as a short-term coping strategy [17]. Nevertheless, in the long term, it can endanger mental and physical health, as it can reduce the subject’s ability to solve problems. Especially in pregnant women, where many changes are occurring, this situation can bring about distress and compromise overall health. Conversely, studies found that emotion-focused coping was the most frequently employed coping strategy reported for women with postpartum depression [42] and it was associated with severe anxiety during pregnancy and after childbirth [43].

Our data shows that women who had problem-focused coping strategies did not present better blood pressure levels than others. However, we found that problem-focused coping has a mediating effect instead of a predictive role. Problem-focused coping strategies are associated with efforts to modify or resolve the stressor and usually help people to experience low emotional stress levels [17], which could lead to healthier blood pressures. However, our search found only one epidemiological study from Stockholm that showed a tendency for higher hypertension rates among subjects scoring low on problem-focused coping [44]. The inconsistency presented by this study may be mainly due to the population accessed. Other studies focus on men and women between 15 and 64 years old, whereas our study just included women older than 18 years. It is also important to consider that pregnancy is an exceptional period. Sudden changes in women’s lives can lead to stress, especially in subjects with hypertension. This scenario can exacerbate the repercussions of emotion-focused coping strategies, such as the experience of negative emotions and stress. This vicious cycle may lead to disorders, including higher blood pressure.

On the other hand, these findings are of interest because it is suggested that women can develop more appropriate coping strategies when facing difficult situations, under some circumstances. In other words, our results point out that achieving higher scores in problem-focused coping strategies in comparison to emotion-focused coping during pregnancy did not influence systolic or diastolic blood pressures. This is a desirable strategy, however, as emotional coping has detrimental effects on blood pressure.

The present study has some limitations. First and foremost, it is an incidental sample and not a random one. Nevertheless, we have a big sample that minimizes selection bias. Data were cross-sectional so the causality among the variables could not be inferred. Moreover, our data excluded women with other comorbidities, such as a previous diagnosis of kidney disease, a history of diabetes, and fetal malformation. These women may present different coping strategies; therefore, studies focused on a better understanding of this situation may be of clinical relevance. Future studies should focus on more complex models and other variables could be included in analysis.

## 5. Conclusions

The computation of the present data establishes that both the openness to experience trait and emotion-focused coping are correlated with systolic and diastolic blood pressure. Detrimental effects on blood pressure were found for emotional coping. Furthermore, problem-focused coping strategies have a mediation effect in personality traits, instead of a predictive role.

Considering the results of the current study, on a theoretical level, it was possible to implement the classic models to better understand the pregnant population, more specifically, women with hypertension. On a practical level, it is very interesting to know that we can improve coping strategies that can mediate the personality effects on blood pressure during pregnancy. Moreover, this not only favors physical health for women and their fetuses, but also their well-being and quality of life.

## Figures and Tables

**Figure 1 healthcare-10-00341-f001:**
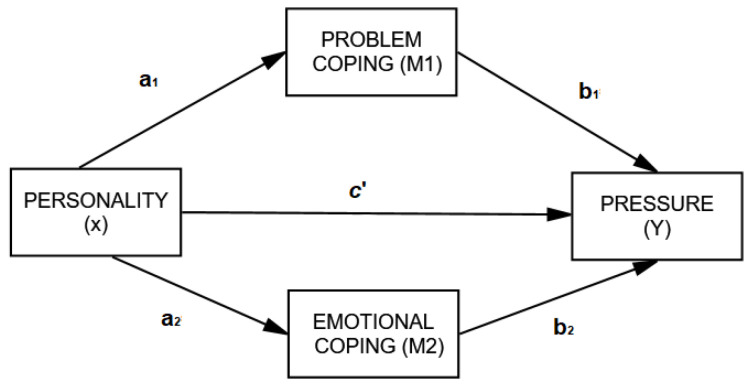
There were no variables with more than 10% missing values. We imputed missing values through the SPSS method for multiple imputations to produce a new data set without missing data. We conducted a mediational analysis using PROCESS macro for SPSS [31] to test the hypothesis that coping mediates the effect of personality on blood pressure. In this way, regression-based procedures were executed by employing bootstrapping procedures using 10,000 samples [32,33].

**Figure 2 healthcare-10-00341-f002:**
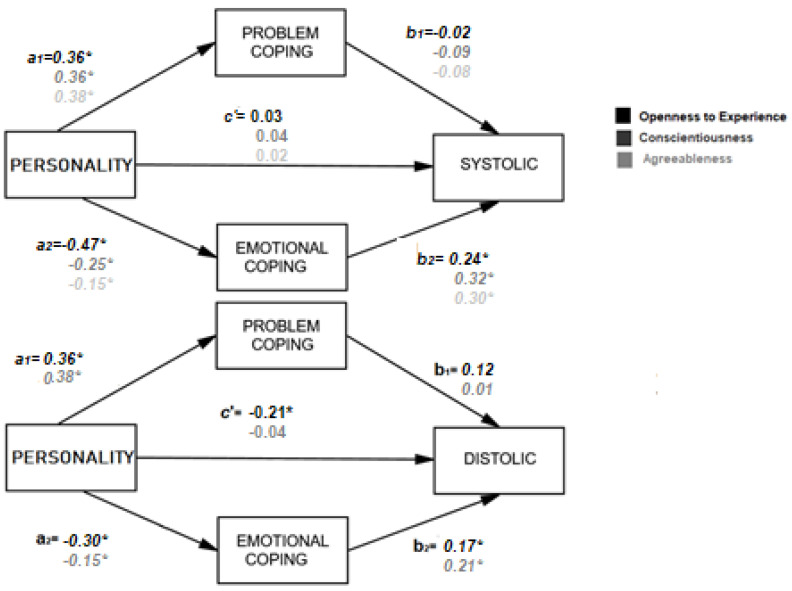
Estimated model, where both strategies of coping moderate the relationship between personality and blood pressure in pregnancy (** = *p* < 0.01; * = *p* < 0.5). This model is based on the first model from Figure 1 which represents a simple statistical mediation model. In the current model, we aim to assess mediation with multiple mediators simultaneously, as coping is considered to be twofold in this model: emotion- and problem-focused (M2 and M1, respectively).

**Table 1 healthcare-10-00341-t001:** Descriptive characteristics of study participants in terms of mean and standard deviation in parenthesis. MA = maternal age; SBP = systolic blood pressure; DBP = diastolic blood pressure; GA = gestational age.

Group	n (%)	MA (Years)	Height (cm)	Weight (kg)	SBP (mmHg)	DBP (mmHg)	GA (Days)
Control	159 (45.3)	26.32 (5.70)	160.71 (6.24)	77.79 (13.60)	118.67 (13.29)	73.94 (11.84)	269.18 (21.90)
Hypertensive	192 (54.7)	28.60 (6.87)	161.68 (6.58)	88.41 (17.62)	146.47 (19.27)	90.20 (13.88)	257.12 (26.07)

**Table 2 healthcare-10-00341-t002:** Pearson coefficients under the variables under study.

	1	2	3	4	5	6	7	8	9
Systolic (1)	1								
Diastolic (2)	0.778 **	1							
Openness to experience (3)	−0.193 **	−0.168 **	1						
Conscientiousness (4)	−0.075	−0.039	0.445 **	1					
Extroversion (5)	−0.128 *	−0.118 *	0.644 **	0.193 **	1				
Agreeableness (6)	−0.056	−0.074	0.357 **	0.344 **	0.210 **	1			
Neuroticism (7)	−0.017	0.013	−0.193 **	−0.049	−0.185 **	−0.524 **	1		
Problem focused coping (8)	0.055	0.088	0.469 **	0.364 **	0.282 **	0.379 **	−0.265 **	1	
Emotion focused coping (9)	0.272 **	0.227 **	−0.044	−0.259 **	−0.009	−0.154 **	−0.046	0.412 **	1

** = *p* < 0.01; * = *p* < 0.05.

**Table 3 healthcare-10-00341-t003:** Effects of X on Y for diastolic blood pressure, standard error (SE), statistical significance, and lower and upper (LLCI and ULCI) levels.

Personality Trait	Effect	Effect	SE	LLCI	ULCI
Openness to experience	Total	−0.17	0.04	−0.26	−0.07
	Direct	−0.21	0.05	−0.32	−0.10
	Total Indirect	0.05	0.03	−0.016	0.115
	a1b1 Indirect	0.05	0.03	−0.0007	0.117
	a2b2 Indirect	0.04	0.02	0.015	0.084
Agreeableness	Total	−0.07	0.05	−0.18	0.03
	Direct	−0.04	0.06	−0.17	0.04
	Total Indirect	−0.02	0.03	−0.09	0.04
	a1b1 Indirect	−0.03	0.01	−0.06	−0.009
	a2b2 Indirect	0.03	0.02	−0.01	0.08

**Table 4 healthcare-10-00341-t004:** Effects of X on Y for systolic blood pressure, standard error (SE), statistical significance, and lower and upper (LLCI and ULCI) levels.

Blood Pressure	Effect	Effect	SE	LLCI	ULCI
openness to experience	Total	−0.04	0.05	−0.176	0.067
	Direct	0.03	0.06	−1.01	0.169
	Total Indirect	−0.073	0.041	−0.158	0.007
	a1b1 Indirect	−0.009	0.026	−0.062	0.041
	a2b2 Indirect	0.052	0.016	0.025	0.092
conscientiousness	Total	−0.07	0.05	−0.177	0.026
	Direct	0.04	0.06	−0.08	0.17
	Total Indirect	−0.11	0.042	−0.20	−0.03
	a1b1 Indirect	−0.03	0.022	−0.084	0.01
	a2b2 Indirect	−0.08	0.02	0.002	0.095
Agreeableness	Total	−0.05	0.05	−0.16	0.04
	Direct	0.02	0.06	−0.10	0.14
	Total Indirect	−0.07	0.03	−0.015	−0.01
	a1b1 Indirect	−0.03	0.02	−0.008	0.01
	a2b2 Indirect	0.04	0.01	−0.09	−0.01

## Data Availability

The data presented in this study are available on request from the corresponding author. The data are not publicly available due to clinical data were investigated and the data protection the participants signed included a statement that the data will not be shared.

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
