# Peer review of "The Mediation Effect of Coping Strategies between Personality and Blood Pressure in Pregnancy Complicated by Hypertension"

_healthcare, 2022, doi:10.3390/healthcare10020341_

Round 1
Reviewer 1 Report
The first author published a similar topic in another journal, as follows.
Chapuis-de-Andrade S, Moret-Tatay C, Quarti Irigaray T, Breno Costa D, Antonello ICF, Pinheiro da Costa BE. (2021). Coping with stress and personality: A study in pregnancies complicated by hypertension. Clin Psychol Psychother. 2021 Nov;28(6):1607-1619. doi: 10.1002/cpp.2603. Epub 2021 May 4. PMID: 33899979.
The current submission looks like a part of finding from above published article.
Author Response
The authors have been working in the field of pregnancy-related hypertension and it’s association with psychological factors such as personality, coping strategies and negative affects for many years. This brief report titled “The mediation effect of coping strategies between personality and blood pressure in pregnancy complicated by hypertension” is part of the Line of Research on Hypertension in Pregnancy, of the Nephrology Research Group from Pontifical Catholic University of Rio Grande do Sul (PUCRS), Porto Alegre, Brazil, which has cooperates internationally with the Universidad Católica de Valencia San Vicente Mártir (UCV), Valencia, Spain and La Sapienza University of Rome.
The current submission belongs to the same project but with different hypotheses, therefore, both methods and results are considerably different in each of the studies.
The study titled “Coping with stress and personality: A study in pregnancies complicated by hypertension”, doi: 10.1002/cpp.2603. Epub 2021 May 4. PMID: 33899979 has investigated the role of personality in pregnancies complicated by hypertension thru analysis of structure and associations between negative affect and coping strategies, and their role towards psychological distress.
To do so, an exploratory approach on the interactions between latent variables was employed, thru a network analysis. This is a graph-theory based methodology which can be used to examine the relationship between observable and latent variables.
On the other hand, our brief report titled “The mediation effect of coping strategies between personality and blood pressure in pregnancy complicated by hypertension” has investigated whether coping strategies can mediate the effect of personality in pregnancies complicated by hypertension.
Moreover, we have upgraded the literature. All references have been revised.
Reviewer 2 Report
Dear Authors,
- In my opinion English editing is needed.
- I think that if you write the references in text the brackets should be square and in brackets should be numbers no names of authors of articles and date of article.
- lines 147-147 I am not sure if I understand the numbers with point in brackets.
Author Response
In my opinion English editing is needed.
We have carefully revised the manuscript.
I think that if you write the references in text the brackets should be square and in brackets should be numbers no names of authors of articles and date of article.
References have been adapted to the Journal style.
lines 147-147 I am not sure if I understand the numbers with point in brackets.
We have reformulated the text to make it clear. The brackets numbers refer to coefficients of internal reliability, Cronbach’s alpha.
Reviewer 3 Report
In the present study, authors investigated the role played by personality and coping strategies in blood pressure levels during pregnancy.
The statistical analysis part needed to be rewritten to clarify what models or statistical tests the authors used for different types of data.
Literature has to be upgraded. The data used by the authors were from the period 2016-2018. On the other side 77% of the references used were older than five years.
Tables were presented in an informal way in the current manuscript. For example, in Table 1. all standard deviation should be added after the mean if it is shown in the items in the next line. Other results were presented very confusingly in the form of tables and were difficult to follow Table 3., Table 4., and Table 5. as well as Figure 2. It might be a good idea to choose another way of the presenting the results.
There were some typos errors.
Author Response
The statistical analysis part needed to be rewritten to clarify what models or statistical tests the authors used for different types of data.
Thank you for the remark. Statistical analysis part has been rewritten, as suggested.
Literature has to be upgraded. The data used by the authors were from the period 2016-2018. On the other side 77% of the references used were older than five years.
Literature has been updated, as suggested.
Tables were presented in an informal way in the current manuscript. For example, in Table 1. all standard deviation should be added after the mean if it is shown in the items in the next line. Other results were presented very confusingly in the form of tables and were difficult to follow Table 3., Table 4., and Table 5. as well as Figure 2. It might be a good idea to choose another way of the presenting the results.
We have reformulated all tables, as suggested.
There were some typos errors.
We have revised the manuscript carefully.
We have highlighted changes in the text in yellow ink. We would like to thank you for all your help and comments to improve the current manuscript.